# Tactic: Adaptive Sparse Attention with Clustering and Distribution Fitting for Long-Context LLMs

**Kan Zhu**[1]  **Tian Tang**[1,2*]  **Qinyu Xu**[1,2*]  **Zhan Jin**[1,3*]  **Yile Gu**[1]  **Zhichen Zeng**[1]
**Rohan Kadekodi**[1]  **Liangyu Zhao**[1]  **Ang Li**[1]  **Arvind Krishnamurthy**[1]  **Baris Kasikci**[1]

[1]University of Washington    [2]Tsinghua University    [3]Peking University

## Abstract

Long-context models are essential for many applications but face inefficiencies in loading large KV caches during decoding. Prior methods enforce fixed token budgets for sparse attention, assuming a pre-chosen number of tokens can approximate full attention. However, these methods overlook variations in the sparsity of attention across heads, layers, and contexts.

To address these limitations, we propose Tactic, an adaptive and calibration-free sparse attention mechanism that dynamically selects tokens based on their cumulative attention scores rather than a fixed token budget. By setting a target fraction of total attention scores, Tactic ensures that token selection naturally adapts to variations in attention sparsity. To efficiently approximate this selection, Tactic leverages clustering-based sorting and distribution fitting, allowing it to accurately estimate token importance with minimal computational overhead.

We show that Tactic achieves superior accuracy and up to $7.29\times$ decode attention speedup, contributing to overall $1.58\times$ end-to-end inference speedup, making it a practical and effective solution for long-context LLM inference in accuracy-sensitive applications.

## 1 Introduction

Large language models (LLMs) power a wide range of applications, from conversational assistants to document analysis systems and search engines. The demand for multi-turn interactions and long-document processing has driven an expansion of context length, growing from thousands to as many as one million tokens (Liu et al., 2024b).

However, supporting long contexts in LLM inference presents significant challenges, primarily due to the growing memory footprint of the Key-Value (KV) cache (Tang et al., 2024; Zhao et al., 2025; 2024). The memory requirements of the KV cache scale proportionally with the context length; therefore, it can quickly become a bottleneck despite optimizations such as Grouped-Query Attention (GQA) (Ainslie et al., 2023). Furthermore, the need to repeatedly load the KV cache for every generated token becomes a bottleneck. For instance, loading the large KV cache can account for over 50% of the total latency during auto-regressive decoding, significantly impeding the efficiency of large-scale serving systems. (Tang et al., 2024; Zhu et al., 2025)

To mitigate the high cost of KV-cache loading, recent methods approximate full attention by selecting a subset of stored Key and Value vectors, corresponding to a subset of tokens, within a fixed token budget (Liu et al., 2024a; Tang et al., 2024; Zhang et al., 2023; Xiao et al., 2023). These approaches exploit the natural sparsity of attention, where only a small fraction of tokens significantly influence the output. By leveraging this sparsity, they aim to reduce the overhead of loading the KV-cache without notably sacrificing model accuracy.

Alas, existing fixed budget-based methods have several shortcomings. Some methods employ a global fixed token budget (Tang et al., 2024; Xiao et al., 2023; Zhang et al., 2023), not accounting for

---

*Work done at University of Washington.

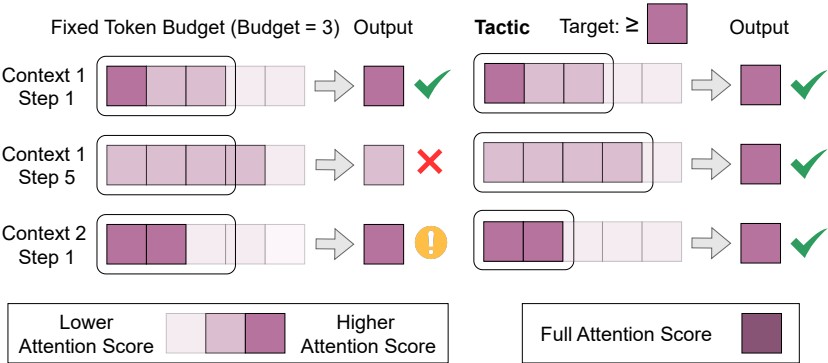

Figure 1: Comparison between fixed-budget-based methods and Tactic. Fixed-budget-based methods may select excessive tokens or have a large difference from full attention score. In contrast, Tactic dynamically selects tokens to efficiently approximate full attention based on a cumulative attention score, considering variation of sparsity across different query tokens and contexts.

variations in attention sparsity across attention heads and layers. In practice, some attention heads focus on significantly more tokens than others, and the level of sparsity fluctuates across layers. More adaptive methods (Cai et al., 2024; Feng et al., 2024; Ge et al., 2024) attempt to distribute token budgets more effectively using calibration data or predefined rules, but they remain constrained by static allocation and cannot adapt to the sparsity variations across query tokens and contexts, often leading to suboptimal approximations in different cases.

To address the limitations of fixed-budget-based methods, we propose Tactic, an adaptive and calibration-free post-training sparse attention mechanism that improves both the accuracy and efficiency of long-context LLM inference. Fig. 1 shows a comparison between existing fixed budget-based methods and Tactic. Instead of enforcing a fixed budget, Tactic dynamically selects tokens starting from ones with the highest attention score (i.e., the softmax output of the Query-Key product) to ensure that their cumulative attention scores reach a target fraction of the full attention score.

Tactic offers key advantages. First, it provides flexibility—Tactic selects fewer tokens in high-sparsity cases and more in low-sparsity cases without requiring calibration. Second, since $V$ vectors have similar norms as we show in Fig. 4, reaching fixed cumulative attention score offers a bounded difference between sparse and full attention (see Sec. 2.3 and Proof 1).

However, efficiently selecting tokens to reach a certain threshold $P$ of cumulative attention score is challenging. To minimize the number of tokens selected (i.e., loads from memory), Tactic ideally needs to select tokens following a descending order of attention score until the cumulative attention score surpasses $P$. Additionally, unlike fixed budget-based methods that simply stop at a fixed token count, Tactic must track cumulative attention score in real time to determine the stop point, making the selection process more complex.

To approximate optimal token selection, Tactic introduces two key techniques: clustering and distribution fitting. First, to efficiently sort tokens, Tactic groups similar tokens to reduce computational overhead. However, we observe that positional proximity, which is used for grouping tokens by prior work (Tang et al., 2024), does not necessarily guarantee similarity in Key vectors, which limits the grouping quality and the attention score estimation. Since attention operates on Query-Key interactions rather than token positions, Tactic groups tokens using K-means clustering based on Key-vector similarity (i.e., vector distance) at the prefill phase. During decoding, Tactic approximates the sorted clusters of tokens based on the similarity between the current Query vector and cluster centroids. After approximating token sorting, Tactic estimates the attention score for each token by leveraging the observation that attention scores follow a smooth distribution. Using distribution fitting, Tactic effectively keeps track of attained cumulative attention score to determine the end of token selection.

By loading only the cluster centroids along with a small sampled subset of tokens ($\sim 2.5\%$ of the KV cache size in practice), Tactic efficiently selects the most critical tokens that reach the target cumulative attention score. To balance efficiency and accuracy, Tactic performs full attention on newly generated tokens and updates the clustering every fixed number of decoding steps (e.g., 2048).

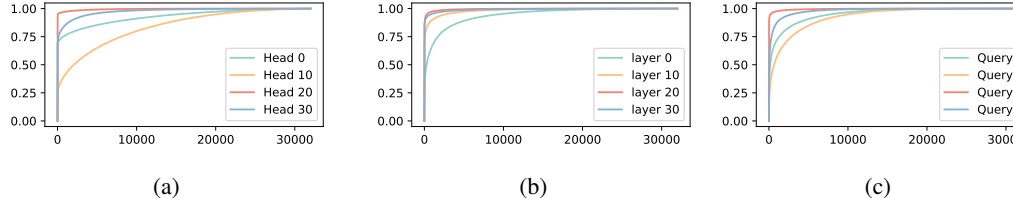

Figure 2: Variation in sparsity across attention heads (a), model layers (b), and query tokens (c). In (c), Query $i$ represents the query vector of $i$-th decoded token. Sorted token indices (by attention score) on the x-axis and the cumulative density of attention score on the y-axis.

Our experiments show that Tactic achieves superior and consistent accuracy compared to existing algorithms, including Quest (Tang et al., 2024), MagicPig (Chen et al., 2024), PyramidKV (Cai et al., 2024), and Ada-KV (Feng et al., 2024), offering a more effective solution for long-context LLM inference in accuracy-sensitive applications. Tactic achieves up to $7.29\times$ decode attention speedup, which leads to $1.58\times$ end-to-end speedup.

In summary, we contribute the following:

- A detailed analysis of the dynamic nature of attention sparsity across heads, layers, queries, and contexts.

- **Tactic**, a sparsity-adaptive attention algorithm that uses clustering and distribution fitting to dynamically determine the token budget for achieving cumulative attention score targets.

- A comprehensive evaluation of Tactic, demonstrating Tactic consistently achieves high accuracy and significant speedup.

## 2 ANALYSIS

### 2.1 INTRINSIC SPARSITY IN SELF-ATTENTION MECHANISMS

In the decode phase, for one request, assuming there are $n$ previous tokens, the attention formula can be written as

$$o = \sum_{i=1}^{n} s_i v_i, \quad s_i = \frac{\exp(\frac{qk_i^\top}{\sqrt{d}})}{\sum_{i=1}^{n} \exp(\frac{qk_i^\top}{\sqrt{d}})}. \tag{1}$$

Our empirical results, evaluated on Llama 3.1 8B model using the PG19 dataset (Rae et al., 2019), show that $\|v_i\|$ are remarkably consistent, while in $s_i$, the exponential term $\exp(\frac{qk_i^\top}{\sqrt{d}})$ non-linearly amplifies the differences in the attention scores, leading to a sparse distribution (Zhang et al., 2023; Xiao et al., 2023). This indicates that a small subset of tokens can exert a significant influence on the model's output, motivating the possibility of only loading a subset of tokens to approximate the attention output and incur lower memory loading overhead.

### 2.2 FIXED TOKEN BUDGET APPROACHES LEAD TO ACCURACY VARIATIONS

Several methods have been proposed to choose a small set of tokens $I$ minimizing the distance $\epsilon(I)$ between full and approximate attention. Some of the work, including Quest (Tang et al., 2024), uniformly chooses tokens across attention heads and layers. These result in a large variance of $\epsilon(I)$, as shown in Fig. 3. This variance stems from the intrinsic sparsity difference across heads and layers. As illustrated in Fig. 2a, attention heads exhibit distinct sparsity patterns. Some heads display a more uniform distribution of $s_i$ (retrieval heads, e.g., head 10), whereas others are dominated by a few high-magnitude $s_i$ values (streaming heads, e.g., head 20). When a fixed number of tokens $|I|$ is selected per head, it leads to inefficiencies—allocating excessive tokens to streaming heads while introducing significant estimation errors in retrieval heads. Similarly, Fig. 2b highlights variation in

$$\epsilon(I) = \left\| o - \frac{1}{p(I)} \sum_{i \in I} s_i v_i \right\|$$
$$= \left\| \sum_{i \in I} s_i v_i + \sum_{i \notin I} s_i v_i - \frac{1}{p(I)} \sum_{i \in I} s_i v_i \right\|$$
$$= \left\| \left(1 - \frac{1}{p(I)}\right) \sum_{i \in I} s_i v_i + \sum_{i \notin I} s_i v_i \right\|$$
$$\leq \left| 1 - \frac{1}{p(I)} \right| \sum_{i \in I} s_i \| v_i \| + \sum_{i \notin I} s_i \| v_i \|$$
$$\leq \left( \frac{1}{p(I)} - 1 \right) p(I) \max_i \| v_i \| + (1 - p(I)) \max_i \| v_i \|$$
$$= 2 \left(1 - p(I)\right) \max_i \| v_i \|. \square$$

Proof 1: Derivation of the error bound.

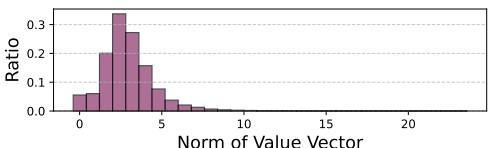

Figure 4: Distribution of $\|V\|$ across layers, heads, and decoding steps. Values concentrate in a narrow range.

sparsity across layers, where layer 0 requires significantly more tokens to reach the same cumulative attention score, making it inefficient to select a fixed number of tokens from different layers.

Motivated by the diversity of sparsity patterns across heads and layers, some works, including AdaKV (Feng et al., 2024) and PyramidKV (Cai et al., 2024), fix the total budget $|I|$ but use calibration data or assumptions to statically assign different budgets to different layers and heads. However, as we show in Fig. 2c, the sparsity of particular heads varies significantly depending on the query token. For example, in the model output *"The Answer is ..."*, the token *"Answer"* attends to far fewer tokens compared to *"is"*. This is because *"Answer"* relies primarily on local information to generate *"is"*, whereas *"is"* requires broader context to produce the subsequent answer. Thus, relying on static partitioning of a fixed token budget also falls short of maintaining a consistent low attention distance $\epsilon(I)$. While MagicPig (Chen et al., 2024) uses a dynamic selection, it does not provide an accuracy guarantee on $\epsilon(I)$.

## 2.3 CUMULATIVE ATTENTION SCORE: A MORE ROBUST TARGET FOR SPARSE ATTENTION

The key drawback of existing work is the reliance on a fixed total token budget, making it hard to adapt to sparsity variations. Instead, we propose directly using the cumulative attention score of tokens in $I$ to guide token selection.

Specifically, we define $p(I)$ as the cumulative attention score of tokens in $I$, which is

$$p(I) = \sum_{i \in I} s_i = \frac{\sum_{i \in I} \exp\left(\frac{q k_i^\top}{\sqrt{d}}\right)}{\sum_{i=1}^{n} \exp\left(\frac{q k_i^\top}{\sqrt{d}}\right)} \quad (2)$$

These cumulative attention score targets offer two key advantages over fixed token budgets. First, they inherently adapt to sparsity variations without requiring assumptions or calibration data. Less sparse heads, layers, query tokens, and contexts naturally require more tokens to reach cumulative attention score than sparser ones.

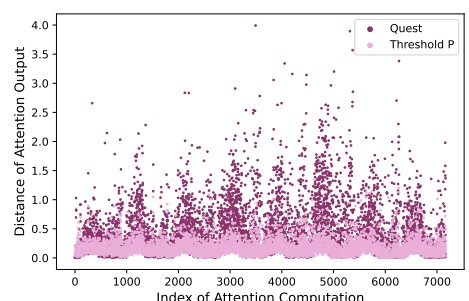

Figure 3: Distance of attention output to full attention of Quest and our proposed solution of selecting tokens until reaching cumulative attention score threshold P, measured with Llama3.1-8B-Instruct model.

Second, targeting cumulative attention score provides a theoretical guarantee on attention distance. Specifically, the attention distance is bounded by $\epsilon(I) \leq 2(1 - p(I)) \max_i \| v_i \|$. Proof 1 provides the full derivation. Since value vectors $V$ have similar norms across tokens (Fig. 4), setting a threshold $P$ (typically close to 1.0) for $p(I)$ establishes a tight upper bound on $\epsilon(I)$. Identifying the minimal index set $I$ that satisfies $p(I) \geq P$ reduces the variance of the attention approximation error, as shown in Fig. 3.

## 2.4 CHALLENGES OF ATTAINING CUMULATIVE ATTENTION SCORES

While effective, identifying the minimal subset of tokens that achieve a target cumulative attention score during the inference is a challenging task. The optimal way is to select tokens following a

Table 1: WCSS of clustering and consecutive grouping, evaluated on Llama-3.1-8B, with data from LongBench (Bai et al., 2024).

| Sequence Length | WCSS (cluster) | WCSS (consecutive) |
|---|---|---|
| 8192 | 173.42 | 195.06 |
| 16384 | 75.29 | 93.29 |
| 32768 | 75.39 | 93.25 |
| 65536 | 78.07 | 93.83 |

descending order of attention score until the cumulative attention score surpasses the target value. Therefore, like prior approaches, Tactic must rank tokens by attention score to minimize the number of tokens needed to reach the desired cumulative attention score. However, unlike previous methods, Tactic also requires the exact attention score values for each token to track the cumulative sum of selected tokens in real-time. This process involves two key components: (1) computing the sum of attention intermediate values, $\exp(qk^\top/\sqrt{d})$, for the selected token set $I$, and (2) computing the total sum of $\exp(qk^\top/\sqrt{d})$ for normalization. Both components need to be effectively approximated without actually calculating the attention score for all tokens. We discuss how Tactic achieves efficient subset identification in the following section.

## 3 METHODOLOGY

### 3.1 ALGORITHM OVERVIEW

Fig. 5 provides an overview of Tactic's workflow. During prefill, Tactic performs K-means clustering on key vectors to group similar tokens. During decode, Tactic ranks tokens based on the dot product between cluster centroids and the current query vector. Tactic then models the distribution of attention score with a fitted curve and determines the tokens to meet the desired cumulative attention score threshold. After token selection, Tactic handles the Group Query Attention (GQA) and then performs the attention using FlashInfer (Ye et al., 2025).

### 3.2 PREFILL STAGE: GROUPING TOKENS VIA CLUSTERING

Similar to prior works, Tactic groups tokens to reduce computational overhead for identifying critical tokens. However, existing methods rely on positional order, assuming consecutive tokens share similar attention patterns (Tang et al., 2024). However, Tab. 1 shows that clustering achieves lower WCSS[1] than consecutive grouping, which means consecutive grouping is suboptimal. Moreover, modern attention kernels efficiently handle non-contiguous KV-cache access, making positional grouping unnecessary. Therefore, Tactic applies K-means clustering to group tokens based on Key-vector similarity during the prefill as a one time cost. Based on the sensitivity analysis in Sec. C.2 on hyper-parameters, Tactic empirically chooses the average cluster size to be 16 to balance accuracy and efficiency and randomly samples initial cluster centroids.[2]

### 3.3 DECODE STAGE: PARTIAL SORT TOKENS USING CLUSTER CENTROIDS

Once the tokens are organized into clusters, Tactic identifies critical clusters for a given query vector $Q$ in the decode phase. The criticality of each cluster is determined by the dot product between $Q$ and each cluster centroid[3]. This process produces a sequence of clusters sorted by the criticality, which can then be unwrapped to obtain a partially sorted token list.

### 3.4 DECODE STAGE: ESTIMATING ATTENTION SCORE VIA DISTRIBUTION FITTING

---

[1] $WCSS = \sum_{x \in C} \|x - \mu_c\|^2$, $\mu_c$ is the mean vector of $C$.

[2] Note that neither multiple initializations nor K-Means ++ initialization drastically improves the clustering quality, and in fact leads to high-performance overhead, as stated in Sec. C.2.

[3] Compared to distance, dot product directly relates to the attention score, which is more accurate.

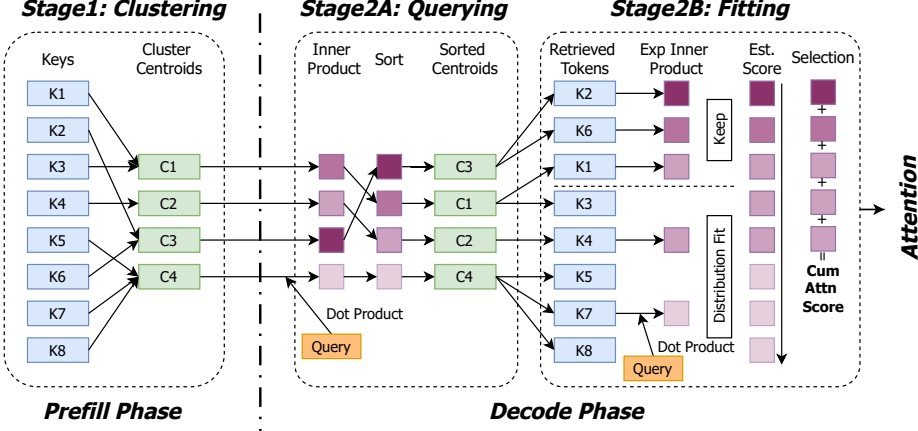

Figure 5: The overall workflow of Tactic. Tactic operates in three stages to achieve low overhead adaptive sparse attention. Stage 1 is applied right after prefill phase, Stage 2A and 2B happens during the decode phase.

Due to the non-linearity of Softmax, cluster centroids do not accurately reflect the average attention score of individual tokens. Thus, Tactic requires a more precise approach to estimating per-token attention score. We observe that after partial sorting, the attention score distribution follows a consistent pattern across heads, layers, and contexts. For example, as shown in Fig. 6, the attention score is high for a few tokens and then smoothly decreases, forming a long-tail distribution suggesting that function fitting can be used to estimate attention score. Tactic models the distribution of the exponential values of the dot products ($\exp(\frac{QK^\top}{\sqrt{d}})$) for each token using a lightweight function $y = \frac{a}{x} + b$, where $x$ is the position index in the sorted list of tokens. To estimate parameters $a$ and $b$, we select two

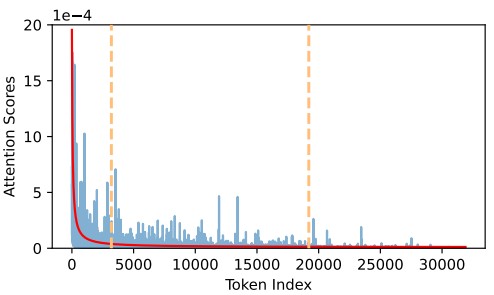

Figure 6: The distribution of attention scores after cluster-based sorting for one request in PG19 dataset using Llama3.1-8B-Instruct model. Despite some variations, the overall trend closely aligns with the function y = $\frac{a}{x} + b$.

segments of the tokens in the middle of the curve (e.g., 10% and 60% of all the tokens), and calculate the average of tokens within each segment (as labeled in Fig. 6). However, initial tokens (1-2% of the total tokens) are often outliers and cannot be accurately described by the curve. Moreover, these tokens feature high attention score, and thus a bad estimation would cause high deviations of estimated cumulative attention score, which affects the accuracy of Tactic. Therefore, Tactic directly calculates the exponential values of the dot products for these tokens. The complete algorithm is provided in Sec. A.

Clustering and distribution fitting operate as a coupled, self-correcting feedback loop that enhances robustness. If the initial token clustering is suboptimal, the resulting attention distribution decays more slowly (i.e., exhibits a flatter curve). Tactic's distribution-fitting procedure detects this flattening and automatically increases the token budget. Moreover, even if the curve-fitting step selects an imperfect cutoff, Tactic's grouping and sorting mechanism continues to prioritize the most critical tokens. Because selection proceeds in priority order, estimation errors mainly shift the cutoff within the low-value tail, leaving high-value tokens largely unaffected.

## 3.5 DECODE STAGE: GQA-AWARE SPARSE ATTENTION

Modern models use Grouped Query Attention (GQA) to reduce the KV cache size (Dubey et al., 2024), where multiple query heads share one KV head. However, existing methods select tokens

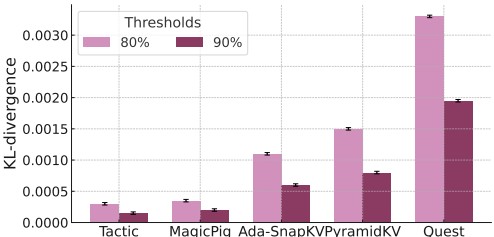 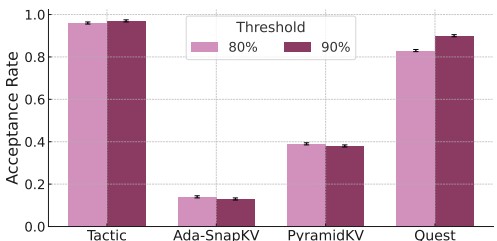

Figure 7: KL-Divergence against full attention of Tactic and other baseline methods on the PG19 dataset. Tactic maintains the most accurate output in two configurations.

Figure 8: Acceptance rate of draft tokens at 80% and 90% thresholds on the PG19 dataset. Tactic shows more than 95% of acceptance rate, surpassing other baselines.

on a per-head basis, and each query head reads KV cache independently. Tactic instead takes the union of selected tokens across all grouped query heads and loads it only once. To ensure workload balancing, Tactic divides each request into subrequests. Each subrequest processes a KV head and its corresponding Query head, with sequence length determined by the tokens selected for each KV head, allowing the request-level workload balance in modern libraries (Ye et al., 2025) to effectively handle head-level imbalance efficiently.

## 4 EXPERIMENTS

### 4.1 SETTING

We evaluate Tactic for both accuracy and efficiency. We mainly use two models: Llama-3.1-8B-Instruct (Grattafiori et al., 2024), a widely used model with Grouped-Query Attention; and MegaBeam-Mistral-7B-512k (Chen Wu and Yin Song and Eden Duthie, 2024), an extended version of Mistral-7B-Instruct-v0.2 with a 512k token context window. To assess cross-model generalization, we additionally evaluate on Qwen2-7B (Yang et al., 2024) and Llama-3.1-70B-Instruct (Grattafiori et al., 2024) (see Sec. B.1).

For accuracy evaluations, we use the PG19 language modeling dataset (Rae et al., 2019), LongBench dataset (Bai et al., 2024), including HotpotQA (Yang et al., 2018), TriviaQA (Joshi et al., 2017), MultifieldQA (Bai et al., 2024), NarrativeQA (Kočiský et al., 2018), Qasper (Dasigi et al., 2021), and Musique (Bai et al., 2024). Additionally, we conduct experiments on the RULER benchmark (Hsieh et al., 2024), using 50 examples for each task. We compare Tactic with the most popular fixed token budget KV cache algorithms, Quest (Tang et al., 2024), PyramidKV (Cai et al., 2024), and Ada-SnapKV (Feng et al., 2024). Also, we compare Tactic with MagicPig (Chen et al., 2024) RULER benchmarks.

For a fair comparison, we first run Tactic to map each target cumulative attention score ratio to a token budget, and then evaluate all baselines under the matched budget. Parameter-wise, we set the page size in Quest and the cluster size in our method to 16. Both Ada-SnapKV and PyramidKV follow the configuration settings outlined in (Feng et al., 2024), including an observation window size of 32 and a max pooling kernel size of 7. We follow the configuration of MagicPig in the original paper in RULER evaluation.

For efficiency evaluations, we perform the evaluation on Nvidia Ada 6000 GPUs with CUDA 12.4 compared with full attention using Flashinfer (Ye et al., 2025).

### 4.2 ACCURACY EVALUATION

#### 4.2.1 OUTPUT ACCURACY

We assess the KL-divergence of the model output probability distribution of Tactic relative to the full attention on the PG19 test set (Rae et al., 2019), under Top-K sampling. We include all texts in PG19 with the number of tokens larger than 32k. In the prefill stage, we truncate the input to 32k tokens

Table 2: Performance comparison on RULER for Llama-3.1-8B-Instruct and Mega-Beam-Mistral-512k models.

*Llama-3.1-8B-Instruct*

| Methods | Config | 16K | 32K | 64K | 96K | Avg. |
|---|---|---|---|---|---|---|
| Full | | 91.3 | 86.0 | 85.2 | 85.0 | 86.8 |
| Tactic | 75% | **90.9** | **85.5** | **83.4** | **78.9** | **84.7** |
| PyramidKV | 75% | 61.8 | 67.4 | 60.8 | 62.5 | 63.1 |
| Ada-SnapKV | 75% | 58.0 | 62.2 | 59.2 | 58.7 | 59.2 |
| Quest | 75% | 70.0 | 71.5 | 69.7 | 65.7 | 69.2 |
| MagicPig | 75% | 78.6 | 76.8 | 70.4 | 70.1 | 74.0 |
| Tactic | 90% | **90.3** | **84.9** | **82.8** | **80.5** | **84.6** |
| PyramidKV | 90% | 73.1 | 76.2 | 74.2 | 68.6 | 73.0 |
| Ada-SnapKV | 90% | 72.7 | 76.4 | 74.3 | 68.7 | 73.0 |
| Quest | 90% | 85.8 | 81.9 | 79.8 | 70.5 | 79.5 |
| MagicPig | 90% | 79.8 | 76.9 | 71.3 | 70.7 | 74.7 |

*Mega-Beam-Mistral-512k*

| Methods | Config | 16K | 32K | 64K | 96K | Avg. |
|---|---|---|---|---|---|---|
| Full | | 90.9 | 88.4 | 82.7 | 83.1 | 86.3 |
| Tactic | 75% | 88.0 | **88.8** | **81.7** | **82.5** | **85.2** |
| PyramidKV | 75% | 80.2 | 79.0 | 75.2 | 74.0 | 77.1 |
| Ada-SnapKV | 75% | 80.6 | 78.1 | 75.4 | 73.6 | 76.8 |
| Quest | 75% | 80.7 | 79.0 | 71.4 | 70.5 | 75.4 |
| MagicPig | 75% | **89.7** | 86.5 | 81.0 | 69.4 | 81.6 |
| Tactic | 90% | **90.3** | **88.0** | 81.0 | 82.6 | **85.4** |
| PyramidKV | 90% | 84.3 | 82.7 | 76.2 | 86.2 | 82.4 |
| Ada-SnapKV | 90% | 84.7 | 81.8 | 76.2 | **87.1** | 82.4 |
| Quest | 90% | 81.1 | 81.3 | 73.5 | 79.7 | 78.9 |
| MagicPig | 90% | 90.2 | 87.1 | 82.3 | 82.3 | 85.4 |

and then feed it into the model. In the decode stage, we use full attention as the reference. We first generate a token sequence with full attention, and then feed the same generated tokens one-by-one into each sparse-attention variant. At each decode step, we collect the output logits and compare them against the corresponding full-attention logits to compute the distributional difference. We collect 32 decode steps in total. As shown in Fig. 7, Tactic achieves the most aligned output compared to all baselines.

### 4.2.2 ACCEPTANCE RATE IN SPECULATIVE DECODING

To further demonstrate the practical indication of smaller KL-divergence, we evaluate the token acceptance rate under greedy sampling when using Tactic as draft model for speculative decoding using the PG19 test set. Specifically, we select all documents in PG19 containing more than 32K tokens and decode up to 96 tokens per document, varying the number of draft tokens (i.e., different values of $\gamma$). During the experiments, we record the verification results, capturing the number of tokens accepted by the target model at each verification step. After computing the average number of accepted tokens for each $\gamma$, we fit a curve to the resulting data points to estimate the acceptance rate, following the formulation in Equation

$$E(\#accepted\ tokens) = \frac{\alpha - \alpha^{\gamma+1}}{1 - \alpha} \qquad (3)$$

which is adapted from (Leviathan et al., 2023). Here $\alpha$ is the acceptance rate to be estimated. The results are presented in Fig. 8. Tactic achieves more than 90% of acceptance rate, significantly surpassing all baselines.

### 4.2.3 ACCURACY FOR LONG-CONTEXT TASKS

**LongBench.** We evaluate Tactic on six LongBench tasks, as illustrated in Sec. 4.1. For each dataset, we first evaluate Tactic by setting the cumulative attention score threshold as 70% and 90%. The average number of tokens selected at each threshold serves as the token budget for evaluating baselines. As shown in Sec. B.2, Tactic beats all other baselines across tasks. At a threshold of 90%, Tactic achieves performance close to full attention. We further validate cross-model and cross-task generalization on Qwen2-7B, Llama-3.1-70B-Instruct, and coding benchmarks in Sec. B.1, where Tactic consistently surpasses all baselines across different architectures, scales, and task domains.

**RULER.** We evaluate Tactic and baselines on all tasks in RULER (Hsieh et al., 2024) with context length ranging from 16K to 96K. As shown in the Tab. 2, Tactic consistently exceeds all baselines in each configuration in terms of average accuracy. Furthermore, at higher thresholds, Tactic achieves similar accuracy to full attention, significantly higher than other methods. Also, we provide the efficiency comparison in the RULER test in Sec. D.1.

We provide a detailed table of the average number of tokens selected by Tactic across various thresholds, datasets, and models in Sec. B.3, which is set as token budgets for baselines.

Table 3: Evaluation of number of tokens selected and ratio of cumulative attention score achieved for Llama-3.1-8B-Instruct model.

| Threshold | Optimal | Cluster Optimal | Tactic | Average Achieved Score | Success Rate |
|---|---|---|---|---|---|
| 50% | 71 | 166 | 185 | 66% | 92% |
| 60% | 122 | 271 | 294 | 72% | 89% |
| 70% | 212 | 451 | 490 | 78% | 86% |
| 80% | 394 | 802 | 890 | 84% | 84% |
| 90% | 895 | 1723 | 1975 | 91% | 86% |

#### 4.2.4 ACCURACY OF CLUSTERING & DISTRIBUTION FITTING

We evaluate our method on the PG19 dataset, focusing on how consistent Tactic can achieve the target cumulative attention score and how many tokens it selects. We vary target cumulative attention score thresholds and compare the number of selected tokens produced by our method against two oracles. The *global optimal* ranks individual tokens by attention score in descending order and selects the smallest prefix whose cumulative attention score meets the target. The *clustering optimal* operates on clusters: it sorts clusters by their aggregate attention score contribution and selects clusters in that order until the target is reached. In addition, we report the realized average cumulative attention score and the success rate, defined as the fraction of cases where the selected tokens achieve at least the target attention score. Tab. 3 indicates that Tactic achieves the target threshold of cumulative attention score with high success rates. Also, the values of *Cluster Optimal* and *Tactic* are close, indicating that the distribution fitting presents an accurate estimation of the number of tokens.

#### 4.2.5 SENSITIVITY STUDY

We analyze the sensitivity of Tactic to its key design choices. First, we compare the inverse fitting function ($y = a/x + b$) against linear, exponential, quadratic, and constant alternatives on Long-Bench (Sec. C.1). All non-constant functions achieve comparable accuracy, but the inverse function consistently selects the fewest tokens, making it the most efficient choice. Additionally, we examine clustering hyperparameters in Sec. C.2: the default configuration (cluster size 16, 10 iterations, single initialization) achieves near-optimal Silhouette scores while minimizing overhead.Finally, we evaluate K-means initialization stability across 8 random seeds in Sec. C.3, where Tactic exhibits strong robustness.

#### 4.2.6 COMPONENT ABLATION

To isolate the contributions of clustering and cumulative attention score–based selection, we introduce two ablated variants on LongBench (Llama-3.1-8B-Instruct): (1) **Tactic-topK**, which retains K-means clustering but uniformly distributes the token budget, and (2) **Position-cluster**, which replaces K-means similarity-based clustering with sequential token grouping. As shown in Sec. C.4, Tactic-topK suffers notable accuracy degradation, while Position-cluster achieves comparable accuracy but requires significantly more tokens.

### 4.3 EFFICIENCY EVALUATION

We begin by analyzing the latency breakdown of Tactic, focusing on token clustering during the prefill phase and attention computation for critical tokens during decoding (Sec. 4.3.1). Next, we evaluate Tactic's end-to-end performance and its speed-up relative to full attention (Sec. 4.3.3).

#### 4.3.1 LATENCY BREAKDOWN

**Latency of clustering during prefill.** We measure the time taken for clustering for different sequence lengths in Fig. 9. We observe that, as the sequence length increases, the clustering time increases quadratically and is dominated by the distance calculation. However, long sequences also significantly increase the prefill time. Overall, the clustering time stays below 7% of the prefill time across all sequence lengths up to 256K. Because both clustering and prefill scale quadratically with sequence length, this ratio remains stable as context grows. For batched inference, clustering can be applied

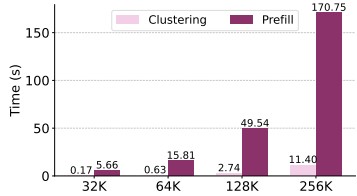

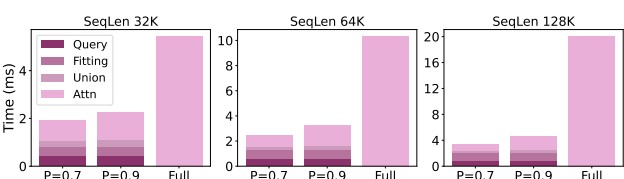

Figure 9: Comparison of Clustering and Prefill time across sequence lengths.

Figure 10: Latency breakdown of Tactic in the decode stage for different sequence lengths and thresholds.

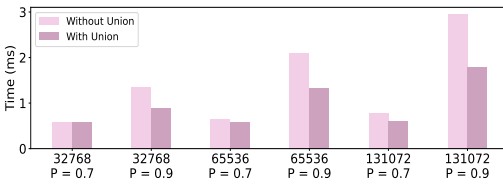

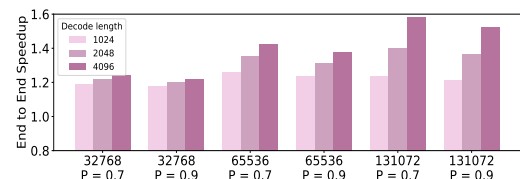

Figure 11: Ablation study on taking union for the GQA model. Taking union significantly reduces the attention time.

Figure 12: End-to-end speedup of Tactic compared to the full attention.

independently per sequence, so both costs scale linearly with batch size, preserving the same overhead ratio.

**Latency of attention during decode.** In the decode stage, Tactic identifies and performs attention on critical tokens. We break down this process into four parts: 1) Cluster sorting, where the clusters are ranked based on the dot product of centroids and queries, 2) Distribution fitting, where Tactic samples a small portion of tokens and derives the attention score to identify the token budget for each attention head, 3) performing attention for the selected tokens. Fig. 10 shows the latency of this breakdown for different sequence lengths.

The latency of sparse attention during decode is reduced significantly, while the overheads of sorting and distribution fitting remain low across various sequence lengths. Overall, Tactic achieves up to $7.29\times$ speedup compared to the full attention.

### 4.3.2 ABLATION STUDY FOR QUERY HEAD UNION

We evaluate the benefits of taking the union of grouped query heads versus computing attention for each query head individually. As shown in Fig. 11, across different context lengths and ratio $P$, taking unions can achieve up to $1.65\times$ attention speedup, due to the reduced memory loading.

### 4.3.3 END-TO-END PERFORMANCE

We compute the end-to-end performance of Tactic with different output tokens, sequence lengths, and ratios in Fig. 12, considering the prefill stages and the clustering overhead. Overall, Tactic achieves a speedup of up to $1.58\times$ compared to full attention.

## 5 CONCLUSION

We presented Tactic, a sparsity-adaptive attention mechanism for efficient long-context LLM inference. Unlike fixed token budget methods, Tactic dynamically selects tokens based on cumulative attention scores, adapting to variations in attention sparsity. By leveraging clustering-based sorting and distribution fitting, Tactic accurately estimates token importance with low overhead. Our results showed that Tactic outperforms existing sparse attention methods, achieving higher accuracy and significant inference speedups, making it a practical solution for long-context LLMs.

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

## A  ALGORITHM OF DISTRIBUTION FITTING

The algorithm for distribution fitting is illustrated in Alg. 1.

---

**Algorithm 1** Estimating Token Budget via Distribution Fitting

---

1: **Input:** Key sequence unpacking from sorted clusters $\{k_1, k_2, \ldots, k_n\}$, query $Q$, weight percentage threshold $P$, initial token count $N$, centers of two sampling window $p_1$ and $p_2$, sampling window size $w$, head dimension $d$
2: **Output:** Token budget $K$
3:
4: Compute $\mu_1$ and $\mu_2$ as the means of $\exp(k_i \cdot Q/\sqrt{d})$ within fixed windows around $p_1$ and $p_2$, namely,

$$\mu_j = \frac{\sum_{i=p_j-\frac{w}{2}}^{p_j+\frac{w}{2}} \exp(k_i \cdot Q/\sqrt{d})}{w}, \ j = 1, 2 \tag{4}$$

Solve for parameters $a$ and $b$ in $y = a/x + b$ using $(p_1, \mu_1)$ and $(p_2, \mu_2)$.
5:
6: Initialize array $s_i$, $i \in [n]$ to store the fitted attention scores for all tokens
7: **for** $i = 1$ to $n$ **do**
8:     If $i \leq N$, $s_i = \exp(x_i \cdot Q/\sqrt{d})$
9:     Else, $s_i = a/i + b$
10: **end for**
11: Compute the minimal $K$ such that the cumulative sum $\sum_1^K s_i \geq P \cdot \sum_1^n s_i$.
12:
13: **return** $K$

---

## B  ADDITIONAL BENCHMARK RESULTS

### B.1  CROSS-MODEL AND CROSS-TASK GENERALIZATION

To assess generalization across model architectures, scales, and task types, we evaluate Tactic on two additional models: Qwen2-7B (Yang et al., 2024) and Llama-3.1-70B-Instruct (Grattafiori et al., 2024) using LongBench tasks. We additionally evaluate on coding benchmarks (RepoBench-P and LCC) from LongBench to test cross-task generalization beyond question answering. As shown in Tab. 4, Tab. 5, and Tab. 6, Tactic consistently achieves near-full-attention accuracy and outperforms all baselines across different models, scales, and task domains.

Table 4: LongBench results on **Qwen2-7B**. Tactic achieves near-full-attention accuracy and consistently outperforms baselines.

| Method | Config | NarrativeQA | HotpotQA | Qasper | MultifieldQA | TriviaQA | Musique | Avg |
|--------|--------|-------------|----------|--------|--------------|----------|---------|-----|
| Full | Full | 24.47 | 36.99 | 45.88 | 47.61 | 85.54 | 20.68 | 43.53 |
| Tactic | 70 | **26.31** | **37.20** | **44.23** | **46.25** | 82.24 | **20.76** | **42.82** |
| PyramidKV | 70 | 20.13 | 35.51 | 25.50 | 39.37 | 82.82 | 20.76 | 37.35 |
| Ada-SnapKV | 70 | 21.91 | 34.86 | 27.58 | 40.83 | **85.81** | 19.23 | 38.37 |
| Quest | 70 | 8.83 | 11.65 | 9.36 | 10.32 | 24.73 | 3.52 | 11.40 |
| Tactic | 90 | **25.71** | **37.26** | **44.08** | **46.19** | 84.26 | **20.79** | **43.05** |
| PyramidKV | 90 | 22.31 | 36.81 | 31.18 | 41.97 | 84.87 | 19.10 | 39.37 |
| Ada-SnapKV | 90 | 21.84 | 35.72 | 34.51 | 44.26 | **86.12** | 21.01 | 40.58 |
| Quest | 90 | 9.97 | 18.46 | 15.10 | 13.64 | 31.78 | 7.02 | 15.99 |

Table 5: LongBench results on **Llama-3.1-70B-Instruct**. Tactic generalizes well to larger model scales.

| Method | Config | HotpotQA | Qasper | MultifieldQA | TriviaQA | Musique | Avg |
|---|---|---|---|---|---|---|---|
| Full | Full | 64.69 | 49.79 | 54.58 | 87.71 | 44.67 | 60.29 |
| Tactic | 70 | **64.71** | **49.64** | **54.28** | 92.16 | **44.51** | **61.06** |
| PyramidKV | 70 | 63.39 | 38.01 | 44.69 | 93.65 | 43.00 | 56.55 |
| Ada-SnapKV | 70 | 63.02 | 35.52 | 50.39 | **93.75** | 42.28 | 56.99 |
| Quest | 70 | 45.91 | 20.12 | 27.84 | 68.40 | 33.37 | 39.13 |
| Tactic | 90 | 64.88 | **49.98** | **54.75** | 89.71 | 44.25 | **60.71** |
| PyramidKV | 90 | 64.08 | 42.14 | 51.25 | **92.98** | 43.96 | 58.88 |
| Ada-SnapKV | 90 | **64.98** | 44.92 | 52.38 | **92.98** | **45.18** | 60.09 |
| Quest | 90 | 54.60 | 42.23 | 46.84 | 78.53 | 39.49 | 52.34 |

Table 6: LongBench **coding** benchmark results on Llama-3.1-8B-Instruct. Tactic achieves near or better accuracy than full attention.

| Method | Config | RepoBench-P | LCC | Avg |
|---|---|---|---|---|
| Full | Full | 52.82 | 49.28 | 51.05 |
| Tactic | 70 | 52.73 | **56.46** | **54.59** |
| PyramidKV | 70 | 51.93 | 49.94 | 50.94 |
| Ada-SnapKV | 70 | 50.80 | 49.45 | 50.12 |
| Quest | 70 | 44.75 | 37.57 | 41.16 |
| Tactic | 90 | 52.51 | 50.73 | 51.62 |
| PyramidKV | 90 | 49.90 | 51.03 | 50.47 |
| Ada-SnapKV | 90 | 51.30 | 50.32 | 50.81 |
| Quest | 90 | **53.71** | **52.80** | **53.25** |

## B.2 LONGBENCH ACCURACY EVALUATION

The LongBench evaluation results are presented in Tab. 7. MagicPig is not included since they don't provide an evaluation script on Longbench.

| Methods | Config | HotpotQA | TriviaQA | MultiFieldQA | Qasper | NarrativeQA | Musique |
|---|---|---|---|---|---|---|---|
| *Llama-3.1-8B-Instruct* | Full | 54.99 | 89.22 | 55.05 | 46.52 | 27.12 | 32.16 |
| Tactic | 70% | 51.20 | **90.38** | **52.98** | **43.30** | **30.39** | **28.57** |
| PyramidKV | 70% | **52.59** | 89.57 | 43.26 | 27.50 | 21.44 | 25.34 |
| AdaKV | 70% | 49.32 | 89.57 | 43.45 | 29.99 | 25.23 | 23.54 |
| Quest | 70% | 45.43 | 77.42 | 49.96 | 38.09 | 24.77 | 24.78 |
| Tactic | 90% | 53.57 | **90.61** | 54.35 | **44.20** | 29.59 | **30.71** |
| PyramidKV | 90% | 53.77 | 90.31 | 48.15 | 36.40 | 26.97 | 28.52 |
| AdaKV | 90% | **54.05** | 90.46 | 49.15 | 37.55 | 27.86 | 29.19 |
| Quest | 90% | 49.37 | 80.38 | 52.42 | 42.41 | **30.21** | 26.64 |
| *MegaBeam-Mistral-512k* | Full | 48.89 | 88.24 | 52.14 | 33.13 | 26.08 | 26.38 |
| Tactic | 70% | **49.15** | 87.89 | 50.50 | **32.37** | **25.63** | **25.85** |
| PyramidKV | 70% | 42.21 | 85.77 | 36.74 | 21.23 | 19.31 | 19.93 |
| AdaKV | 70% | 42.23 | 85.65 | 38.44 | 22.23 | 21.89 | 21.68 |
| Quest | 70% | 48.90 | **88.13** | 50.58 | 30.78 | 23.88 | 24.65 |
| Tactic | 90% | 49.59 | **89.16** | 49.85 | **33.93** | **26.31** | 25.93 |
| PyramidKV | 90% | 44.05 | 86.64 | 40.66 | 24.22 | 21.13 | 23.32 |
| AdaKV | 90% | 44.80 | 86.80 | 42.80 | 22.51 | 22.46 | 24.86 |
| Quest | 90% | **51.69** | 88.49 | **51.81** | 32.46 | 24.63 | 25.89 |

Table 7: Experiment Results on LongBench

## B.3 NUMBER OF TOKENS SELECTED BY TACTIC IN BENCHMARKS

We provide the average number of tokens selected by Tactic in benchmark evaluations in Tab. 8, Tab. 9 and Tab. 10.

Table 8: Average number of tokens selected by Tactic for different cumulative attention scores.

| P | HotpotQA | TriviaQA | MultiFieldQA | Qasper | NarrativeQA | Musique |
|---|---|---|---|---|---|---|
| | | | *Llama-3.1-8B-Instruct* | | | |
| 70% | 959 | 761 | 774 | 629 | 1918 | 1229 |
| 90% | 2298 | 1813 | 1559 | 1276 | 4254 | 2754 |
| | | | *MegaBeam-Mistral-7B-512k* | | | |
| 70% | 2126 | 1733 | 1399 | 1191 | 3017 | 2598 |
| 90% | 3641 | 3048 | 2031 | 1546 | 5616 | 4384 |

Table 9: Average number of tokens selected by Tactic for Llama-3.1-8B-Instruct across context lengths and cumulative attention scores.

| Task | 16K | | 32K | | 64K | | 96K | |
|---|---|---|---|---|---|---|---|---|
| | 75% | 90% | 75% | 90% | 75% | 90% | 75% | 90% |
| NIAH_Single 1 | 166 | 813 | 363 | 1567 | 456 | 2404 | 1790 | 3319 |
| NIAH_Single 2 | 271 | 1289 | 534 | 2196 | 711 | 3209 | 2030 | 3940 |
| NIAH_Single 3 | 171 | 1015 | 369 | 1820 | 507 | 3031 | 1068 | 3952 |
| NIAH_Multikey 1 | 224 | 1052 | 449 | 1832 | 654 | 2792 | 978 | 4438 |
| NIAH_Multikey 2 | 399 | 1612 | 706 | 2533 | 981 | 3902 | 1405 | 5527 |
| NIAH_Multikey 3 | 340 | 1428 | 567 | 2404 | 798 | 3990 | 1068 | 5062 |
| NIAH_Multivalue | 246 | 1769 | 478 | 2679 | 692 | 4458 | 1025 | 8147 |
| NIAH_Multiquery | 253 | 1524 | 465 | 2648 | 681 | 4830 | 915 | 6011 |
| FWE | 282 | 1572 | 515 | 2693 | 778 | 4376 | 963 | 6202 |
| CWE | 443 | 1939 | 570 | 3036 | 655 | 4707 | 780 | 6731 |
| QA 1 | 329 | 1250 | 704 | 2565 | 852 | 3830 | 3417 | 5055 |
| QA 2 | 547 | 1484 | 1092 | 2263 | 1662 | 3455 | 2120 | 4276 |
| VT | 118 | 731 | 253 | 1413 | 269 | 2028 | 404 | 3068 |
| AVG | 291 | 1344 | 543 | 2281 | 746 | 3616 | 1382 | 5056 |
| AVG/Token | 1.78% | 8.21% | 1.66% | 6.96% | 1.14% | 5.52% | 1.41% | 5.14% |

Table 10: Average number of tokens selected by Tactic for MegaBeam-Mistral-7B-512K across context lengths and cumulative attention scores.

| Task | 16K | | 32K | | 64K | | 96K | |
|---|---|---|---|---|---|---|---|---|
| | 75% | 90% | 75% | 90% | 75% | 90% | 75% | 90% |
| NIAH_Single 1 | 1410 | 1176 | 2521 | 2354 | 4715 | 4512 | 7042 | 7271 |
| NIAH_Single 2 | 1418 | 1665 | 2704 | 3364 | 5113 | 6732 | 7668 | 10472 |
| NIAH_Single 3 | 3005 | 2032 | 5312 | 3800 | 10235 | 7327 | 15492 | 11090 |
| NIAH_Multikey 1 | 1486 | 1661 | 2872 | 3415 | 5332 | 5983 | 7781 | 8767 |
| NIAH_Multikey 2 | 1480 | 1639 | 2826 | 3031 | 5288 | 5956 | 8240 | 9279 |
| NIAH_Multikey 3 | 2840 | 2039 | 5990 | 4306 | 11438 | 7741 | 17088 | 11292 |
| NIAH_Multivalue | 2880 | 2225 | 5070 | 4006 | 10180 | 7463 | 16138 | 11853 |
| NIAH_Multiquery | 3088 | 2165 | 5234 | 4075 | 9931 | 7551 | 14420 | 11164 |
| FWE | 2706 | 1809 | 5537 | 3685 | 10837 | 7157 | 16558 | 10818 |
| CWE | 4240 | 2526 | 7404 | 4869 | 13189 | 8802 | 18329 | 13215 |
| QA 1 | 1003 | 1502 | 2772 | 3132 | 5828 | 5685 | 6007 | 9150 |
| QA 2 | 724 | 1718 | 2124 | 3017 | 4853 | 6520 | 8598 | 8598 |
| VT | 2227 | 1409 | 4132 | 2619 | 8667 | 4719 | 14318 | 8671 |
| AVG | 2193 | 1813 | 4192 | 3513 | 8124 | 6627 | 12129 | 10126 |
| AVG/Token | 13.38% | 11.06% | 12.79% | 10.72% | 12.40% | 10.11% | 12.34% | 10.30% |

## C  ABLATION AND SENSITIVITY STUDIES

### C.1  DISTRIBUTION FITTING ALGORITHM

We compare five curve fitting algorithms for estimating the attention score distribution: inverse ($y = a/x + b$), linear ($y = ax + b$), exponential ($y = ae^{bx}$), quadratic ($y = ax^2 + bx + c$), and

constant (uniform budget). Tab. 11 reports accuracy and average number of tokens selected on six LongBench tasks. All non-constant curves model the monotonically decreasing trend of attention scores and achieve comparable accuracy. Among them, the inverse function consistently selects the fewest tokens while maintaining high accuracy, making it the most efficient choice.

Table 11: Ablation study on distribution fitting algorithms. We report accuracy and average number of tokens selected (Tok) on LongBench tasks at cumulative attention score thresholds of 70% and 90%.

| Fit Algo | Thr. | HotpotQA | Tok | MultifieldQA | Tok | Musique | Tok | NarrativeQA | Tok | Qasper | Tok | TriviaQA | Tok |
|---|---|---|---|---|---|---|---|---|---|---|---|---|---|
| Inverse | 70 | **56.89** | **719** | 52.81 | **459** | 29.87 | **878** | 29.34 | **1417** | 44.74 | 336 | 89.46 | **599** |
| Linear | 70 | 55.90 | 764 | 52.96 | 499 | 29.67 | 944 | 29.28 | 1465 | 45.28 | 377 | 88.71 | 620 |
| Exp | 70 | **56.89** | 761 | **53.49** | 496 | **30.09** | 928 | 29.35 | 1501 | 45.15 | 369 | 89.16 | 621 |
| Quadratic | 70 | 56.17 | 776 | 52.98 | 508 | 29.88 | 953 | **29.85** | 1518 | 44.66 | 379 | 89.76 | 636 |
| Const | 70 | 55.84 | 11072 | 52.46 | 6561 | 27.81 | 13236 | 29.05 | 20010 | **45.49** | 4393 | **90.26** | 9852 |
| Inverse | 90 | **57.17** | **1707** | 54.19 | **1251** | **29.57** | **2056** | 29.41 | **3460** | 44.69 | **990** | 90.30 | **1356** |
| Linear | 90 | 56.67 | 1812 | 53.52 | 1335 | 29.54 | 2203 | **30.38** | 3740 | 44.80 | 1051 | **90.80** | 1397 |
| Exp | 90 | 56.67 | 1769 | **54.26** | 1293 | 29.46 | 2142 | 29.43 | 3603 | 45.28 | 1003 | **90.80** | 1381 |
| Quadratic | 90 | 56.67 | 1884 | 53.74 | 1387 | 29.54 | 2287 | 30.18 | 3863 | 44.69 | 1099 | 90.40 | 1463 |
| Const | 90 | 56.67 | 12661 | 52.46 | 7478 | 28.21 | 15148 | 29.25 | 23011 | **45.32** | 4987 | 90.26 | 11319 |

## C.2 SENSITIVITY ANALYSIS OF CLUSTERING

As stated in Sec. 3.2, clustering is controlled by three parameters: the number of iterations, the cluster size, and the initialization count (n_init). In this section we provide the effects of different value of the three parameters. We use Silhouette Score (Rousseeuw, 1987) to evaluate the performance of clustering, which is a commonly used metric in machine learning field.

Tab. 12 presents the impact of key hyperparameters on clustering quality across different sequence lengths. Increasing the number of iterations or number of initializations brings little improvement on the performance of clustering but proportional overhead, Tactic chooses 10 iterations and one time initialization to maintain high performance with minimal overhead. Also, a smaller cluster size yields better clustering performance but longer clustering time, Tactic chooses 16 to balance the performance and efficiency. Notably, the effect of these hyperparameters is largely *independent of sequence length*.

| Hyperparameter | Value | 8192 | 16384 | 32768 | 65536 |
|---|---|---|---|---|---|
| **n_init** | 1 | 0.091 | 0.113 | 0.145 | 0.166 |
| | 2 | 0.095 | 0.117 | 0.150 | 0.170 |
| | 4 | 0.095 | 0.117 | 0.149 | 0.170 |
| | 8 | 0.095 | 0.118 | 0.149 | 0.170 |
| **max_iter** | 5 | 0.091 | 0.111 | 0.145 | 0.166 |
| | 10 | 0.091 | 0.113 | 0.144 | 0.165 |
| | 20 | 0.095 | 0.117 | 0.150 | 0.171 |
| | 30 | 0.095 | 0.117 | 0.150 | 0.171 |
| | 40 | 0.095 | 0.117 | 0.150 | 0.171 |
| | 50 | 0.095 | 0.117 | 0.150 | 0.171 |
| **cluster_size** | 8 | 0.096 | 0.118 | 0.155 | 0.173 |
| | 16 | 0.095 | 0.116 | 0.149 | 0.170 |
| | 32 | 0.088 | 0.110 | 0.143 | 0.164 |
| | 64 | 0.082 | 0.104 | 0.131 | 0.155 |
| | 128 | 0.070 | 0.096 | 0.119 | 0.143 |

Table 12: Silhouette scores for K-Means clustering across different sequence lengths, evaluated against hyperparameters **n_init**, **max_iter**, and **cluster_size**.

## C.3 RANDOM SEED STABILITY

To verify the robustness of Tactic's clustering-based approach, we evaluate on LongBench (Llama-3.1-8B-Instruct, threshold 90%) using eight random seeds (100, 200, . . . , 800) for K-means initialization.

As shown in Tab. 13, Tactic exhibits strong stability across runs, with a standard deviation of approximately 0.2 per task and only 0.05 on the overall average. This confirms that the large number of clusters and data points per dataset effectively mitigates the randomness of K-means initialization.

Table 13: LongBench accuracy across 8 random seeds (threshold 90%). Tactic exhibits strong stability with std $\approx 0.2$.

| Seed | HotpotQA | MultifieldQA | Musique | NarrativeQA | Qasper | TriviaQA | Avg |
|---|---|---|---|---|---|---|---|
| 100 | 56.52 | 54.15 | 28.90 | 29.35 | 44.70 | 90.80 | 50.74 |
| 200 | 57.17 | 54.14 | 29.13 | 29.35 | 44.47 | 90.80 | 50.84 |
| 300 | 57.17 | 54.11 | 29.13 | 29.27 | 44.76 | 90.30 | 50.79 |
| 400 | 57.17 | 54.24 | 28.67 | 29.23 | 44.80 | 90.80 | 50.82 |
| 500 | 56.92 | 53.81 | 29.13 | 29.41 | 45.02 | 90.80 | 50.85 |
| 600 | 57.17 | 53.77 | 29.13 | 29.61 | 44.77 | 90.80 | 50.88 |
| 700 | 56.92 | 54.26 | 29.13 | 29.85 | 44.56 | 90.80 | 50.92 |
| 800 | 57.02 | 53.77 | 29.13 | 29.36 | 44.65 | 90.80 | 50.79 |
| Std | 0.21 | 0.20 | 0.16 | 0.19 | 0.16 | 0.17 | 0.05 |

## C.4 COMPONENT ABLATION

To isolate the contributions of clustering and cumulative attention score–based selection in Tactic, we compare the full system against two ablated variants on LongBench (Llama-3.1-8B-Instruct) at cumulative attention score thresholds of 70% and 90%:

- **Tactic (full)**: K-means clustering groups tokens by key-vector similarity; clusters are ranked by attention score and the token budget is adaptively determined via distribution fitting.
- **Tactic-topK**: Retains K-means clustering but uniformly distributes the token budget (as determined by Tactic) across all layers and heads, ignoring the adaptive scoring mechanism.
- **Position-cluster**: Replaces K-means similarity-based clustering with sequential token grouping (consecutive windows), while keeping the distribution fitting pipeline unchanged.

As shown in Tab. 14, Tactic-topK fails to allocate tokens effectively to critical layers and heads, leading to notable accuracy degradation (e.g., HotpotQA drops from 57.39 to 43.38 at 70% threshold). Position-cluster yields lower-quality clusters, requiring more tokens to achieve the same cumulative attention score target (e.g., 33% more tokens on average at 70%). However, since the cumulative score target is fixed, Position-cluster ultimately achieves similar accuracy but at a higher memory loading cost. These results confirm that both high-quality clustering and score-based token selection are essential for Tactic's performance-efficiency tradeoff.

Table 14: Component ablation on LongBench (Llama-3.1-8B-Instruct). We report accuracy (Acc) and average number of tokens selected (Tok) at cumulative attention score thresholds of 70% and 90%.

| Variant | Thr. | HotpotQA | Tok | MultifieldQA | Tok | Musique | Tok | NarrativeQA | Tok | Qasper | Tok | TriviaQA | Tok |
|---|---|---|---|---|---|---|---|---|---|---|---|---|---|
| Tactic | 70 | **57.39** | 719 | **52.71** | 458 | **29.87** | 877 | 29.27 | 1413 | 45.22 | 334 | **89.46** | 596 |
| Tactic-topK | 70 | 43.38 | 719 | 47.53 | 458 | 16.47 | 877 | 29.82 | 1413 | 39.00 | 334 | 50.49 | 596 |
| Position-cluster | 70 | 55.62 | 908 | 52.29 | 589 | 28.76 | 1102 | **30.90** | 2053 | 45.05 | 444 | 89.71 | 752 |
| Tactic | 90 | **57.02** | 1704 | **54.16** | 1252 | **28.96** | 2072 | 29.41 | 3429 | 44.68 | 985 | **90.80** | 1356 |
| Tactic-topK | 90 | 52.74 | 1704 | 52.18 | 1251 | 27.29 | 2072 | 29.40 | 3429 | 44.60 | 985 | 73.80 | 1356 |
| Position-cluster | 90 | 56.52 | 2398 | 52.94 | 1652 | 27.54 | 2828 | **30.57** | 5550 | **45.39** | 1290 | 90.13 | 1936 |

## D ADDITIONAL EFFICIENCY RESULTS

### D.1 SPEEDUP OF TACTIC IN RULER EVALUATION

To ensure a fair comparison of efficiency between Tactic and baseline methods in Sec. 4.2.3, we adjust the number of selected tokens to match the RULER scores. The token count required to achieve this accuracy, along with the attention speedup over baseline methods, is shown in Tab. 15.

Table 15: Attention speedup over baselines in RULER evaluation.

| Method | SeqLen | # Tokens Chosen |
|---|---|---|
| Tactic | 32K | 543 (1×) |
| Quest | 32K | 5430 (10×) |
| Pyramid | 32K | 8145 (15×) |
| Ada | 32K | 8688 (16×) |
| Tactic | 64K | 746 (1×) |
| Quest | 64K | 9698 (13×) |
| Pyramid | 64K | 16412 (22×) |
| Ada | 64K | 14920 (20×) |
| Tactic | 128K | 1381 (1×) |
| Quest | 128K | 20715 (15×) |
| Pyramid | 128K | 34525 (25×) |
| Ada | 128K | 33144 (24×) |

# E  COMPUTER INFORMATION OF EXPERIMENTS

For Longbench and RULER evaluation (Tab. 7 and Tab. 2), we use a Nvidia H100 DGX server, Longbench evaluation takes around 8 hours and RULER evaluation takes 12 hours, two models (Llama-3.1-8B-Instruct and MegaBeam-Mistral-7B-512K). Other accuracy evaluations like KL-divergence can be done within 30 minutes on a single H100. For Efficiency evaluation, we use Nvidia Ada 6000 GPUs with CUDA 12.4.

# F  POTENTIAL SOCIAL IMPACTS

This paper is motivated by recent advances in the field of long-context language models. Sparse attention methods has the potential to be used to serve more requests and thus benefit more users.

