# OpenReview forum: "Tactic: Adaptive Sparse Attention with Clustering and Distribution Fitting for Long-Context LLMs"
_ICLR.cc/2026/Conference — ICLR 2026 Poster_

### Official Review · Reviewer_gwJo · 2025-10-29

**Soundness:** 3
**Presentation:** 3
**Contribution:** 3
**Rating:** 6
**Confidence:** 4

**Summary:**

This paper introduces Tactic, an adaptive and calibration-free sparse attention mechanism for long-context LLMs. Unlike prior methods that use a fixed token budget, Tactic dynamically selects tokens by targeting a fraction of the total attention scores, allowing it to naturally adapt to sparsity variations across heads and layers. To achieve this efficiently, it leverages clustering-based sorting and distribution fitting. The method reports significant improvements, including up to a 7.29× speedup in decode attention and a 1.58× end-to-end inference speedup while maintaining accuracy.

**Strengths:**

1.  The core idea of targeting a fraction of attention scores instead of a fixed budget is elegantly designed to address the limitations of fixed-budget methods in handling variable attention sparsity.
2.  The method is calibration-free, which significantly lowers the barrier for practical adoption and simplifies its integration into existing inference pipelines.
3.  The combination of clustering and distribution fitting appears to be a computationally efficient approach to enable the adaptive selection mechanism.
4.  The reported performance gains are substantial, suggesting that Tactic is a promising solution for accelerating long-context LLM inference.

**Weaknesses:**

1.  The method has two main components, clustering and cumulative attention score based selection. It's not clear how the two components work together to achieve the accuracy gains. An ablation study is needed to isolate the contributions of the two components.

**Questions:**

1.  Could you provide an ablation study that isolates the contributions of the clustering-based sorting, the distribution fitting components and the cumulative attention score based selection to the overall performance?

---

> ### Author Response · Authors · 2025-11-26
> **Thanks for your insightful and thoughtful review. We would like to address these concerns in the following sections.**
>
> ## How Do Clustering And Score-Based Selection Each Contribute To Tactic’s Accuracy Gains?
>
> We include an ablation study to isolate the contributions of clustering and cumulative attention score–based selection by introducing two baselines:
>
> - **Tactic-topK**: This baseline retains the clustering component but uniformly distributes the token budget (as determined by Tactic) across all layers and heads, ignoring the adaptive scoring mechanism.
>
> - **Position-cluster**: This uses trivial clustering, where sequential tokens are grouped into clusters instead of using attention-based similarity.
>
> Tactic-topK fails to allocate tokens effectively to critical layers and heads, leading to **notable accuracy degradation**. On the other hand, Position-cluster yields lower-quality clusters, **requiring more tokens to achieve the cumulative attention score**. However, since the cumulative attention score target is fixed, it ultimately achieves similar results, but at a higher memory loading cost.
>
> These results highlight that both components, high-quality clustering and score-based token selection, are essential for Tactic's effectiveness. Neither alone can match the performance-efficiency tradeoff achieved by the full method.
>
>
> Variant|Threshold|Hotpotqa Acc|Hotpotqa Tok|Multifieldqa_en Acc|Multifieldqa_en Tok|Musique Acc|Musique Tok
> ---|---|---|---|---|---|---|---
> Tactic|70|**57.39**|**719**|**52.71**|**458**|**29.87**|**877**
> Tactic_topK|70|43.38|719|47.53|458|16.47|877
> Position_cluster|70|55.62|908|52.29|589|28.76|1102
> |||||||
> Tactic|90|**57.02**|**1704**|**54.16**|**1252**|**28.96**|**2072**
> Tactic_topK|90|52.74|1704|52.18|1251|27.29|2072
> Position_cluster|90|56.52|2398|52.94|1652|27.54|2828
>
> Variant|Threshold|Narrativeqa Acc|Narrativeqa Tok|Qasper Acc|Qasper Tok|Triviaqa Acc|Triviaqa Tok
> ---|---|---|---|---|---|---|---
> Tactic|70|29.27|**1413**|**45.22**|**334**|**89.46**|**596**
> Tactic_topK|70|29.82|1413|39.00|334|50.49|596
> Position_cluster|70|**30.90**|2053|45.05|444|89.71|752
> |||||||
> Tactic|90|29.41|**3429**|44.68|**985**|**90.80**|**1356**
> Tactic_topK|90|29.40|3429|44.60|985|73.80|1356
> Position_cluster|90|**30.57**|5550|**45.39**|1290|90.13|1936

---

### Official Review · Reviewer_tuQx · 2025-10-30

**Soundness:** 3
**Presentation:** 3
**Contribution:** 2
**Rating:** 6
**Confidence:** 4

**Summary:**

Paper points out that using a fixed number of tokens for attention is inefficient because different inputs require appropriate or different amounts of focus. To solve this, TACTIC adjusts the number of tokens dynamically—it keeps only as many tokens as needed to capture most of the important information in the attention layer. It does this by grouping similar tokens with K-means clustering during setup and by modeling how attention changes during generation. This method makes the model run much faster (7× faster in attention calculations) while keeping accuracy almost the same.

**Strengths:**

It is a smart and adaptive approach to improving efficiency in large language models. Instead of using a fixed number of tokens for attention, which can waste computation on unimportant parts it dynamically selects only the most relevant tokens using clustering and distribution fitting. This makes the smaller model(8~9B)  much faster while keeping accuracy almost unchanged, and it’s done without extra training or calibration, making it easy to apply to existing models.

**Weaknesses:**

This method mainly focuses on the decode phase, so, it doesn’t speed up the initial setup (prefill) as much. Additionally, although the results are strong, the paper doesn’t thoroughly examine how the method performs on large models(e.g., 70B and above) or various types of tasks, and it lacks detailed statistical analysis across multiple runs to confirm consistency.

**Questions:**

1. Since the paper focuses on a few model families, is the adaptive token selection strategy architecture-agnostic, as it would support broader applicability?

2. Can the authors provide results across multiple random seeds or include confidence intervals to confirm reproducibility and eliminate the possibility that observed improvements are dataset or run-specific?

3. Has there been consideration to compare against the very latest “heavy hitter / speculative decode / KV eviction + reuse” on real hardware latency, not just attention microbenchmarks?

---

> ### Author Response · Authors · 2025-11-26
> **Thanks for your insightful and thoughtful review. We would like to address these concerns in the following sections.**
>
> ## Why Does Tactic Primarily Accelerate The Decode Phase Rather Than Prefill?
>
> Prefill and decode have fundamentally different computational characteristics. During decoding, Tactic only needs to evaluate the importance of a small number of critical tokens for each newly generated token. In contrast, the prefill phase requires reasoning about pairwise relationships across the full input sequence, which requires a different methodology.
>
> While Tactic does not accelerate the prefill phase, prefill‑sparse attention has been extensively explored in prior work, and those methods can be applied orthogonally alongside Tactic. In practice, **this makes Tactic complementary rather than overlapping with existing prefill optimizations**.
>
> Moreover, **decode-heavy workloads are increasingly common** (e.g., mathematical/chain-of-thought reasoning, code generation, agentic workflows, where the decode length can exceed 32K tokens), making decode‑phase acceleration particularly valuable. As shown in Figure 11, even without any prefill optimization, Tactic already delivers up to **1.58× end-to-end speedup**. This benefit would further increase for workloads with even longer decode sequences.
>
>
>
> ## How Consistent Are Tactic’s Improvements Across Random Seeds And Datasets?
>
> We evaluate Tactic using multiple random seeds: 100, 200, ..., 800. Thanks to the large sample size per dataset and the high number of clusters, **the method exhibits strong stability across runs, with a standard deviation around 0.2**, suggesting that Tactic's gains are robust.
>
> Seed|Hotpotqa|Multifieldqa_en|Musique|Narrativeqa|Qasper|Triviaqa|Avg
> -|-|-|-|-|-|-|-
> 100|56.52|54.15|28.90|29.35|44.70|90.80|50.74
> 200|57.17|54.14|29.13|29.35|44.47|90.80|50.84
> 300|57.17|54.11|29.13|29.27|44.76|90.30|50.79
> 400|57.17|54.24|28.67|29.23|44.80|90.80|50.82
> 500|56.92|53.81|29.13|29.41|45.02|90.80|50.85
> 600|57.17|53.77|29.13|29.61|44.77|90.80|50.88
> 700|56.92|54.26|29.13|29.85|44.56|90.80|50.92
> 800|57.02|53.77|29.13|29.36|44.65|90.80|50.79
> ||
> **Std**|0.21|0.20|0.16|0.19|0.16|0.17|0.05
>
>
> ## How Does Tactic Perform On Other Models And Diverse Task Types?
>
> We provide Longbench evaluation results on two additional models, Qwen2 and Llama-3.1-70B-Instruct, to assess generalizability. In both cases, Tactic achieves near-full attention accuracy, consistently outperforming baseline methods. These results suggest that Tactic generalizes well across different model architectures and scales.
>
> ### Qwen2 7B:
> Method|Config|Narrativeqa|Hotpotqa|Qasper|Multifieldqa En|Triviaqa|Musique|Average
> -|-|-|-|-|-|-|-|-
> Full|Full|24.47|36.99|45.88|47.61|85.54|20.68|43.53
> Tactic|70|**26.31**|**37.20**|**44.23**|**46.25**|82.24|**20.76**|**42.82**
> PyramidKV|70|20.13|35.51|25.50|39.37|82.82|20.76|37.35
> AdaKV|70|21.91|34.86|27.58|40.83|**85.81**|19.23|38.37
> Quest|70|8.83|11.65|9.36|10.32|24.73|3.52|11.40
> ||
> Tactic|90|**25.71**|**37.26**|**44.08**|**46.19**|84.26|**20.79**|**43.05**
> PyramidKV|90|22.31|36.81|31.18|41.97|84.87|19.10|39.37
> AdaKV|90|21.84|35.72|34.51|44.26|**86.12**|**21.01**|40.58
> Quest|90|9.97|18.46|15.10|13.64|31.78|7.02|15.99
>
> ### Llama-3.1-70B-Instruct:
> Method|Config|Hotpotqa|Qasper|Multifieldqa En|Triviaqa|Musique|Average
> -|-|-|-|-|-|-|-
> Full|Full|64.69|49.79|54.58|87.71|44.67|60.29
> Tactic|70|**64.71**|**49.64**|**54.28**|92.16|**44.51**|**61.06**
> PyramidKV|70|63.39|38.01|44.69|93.65|43.00|56.55
> AdaKV|70|63.02|35.52|50.39|**93.75**|42.28|56.99
> Quest|70|45.91|20.12|27.84|68.40|33.37|39.13
> ||
> Tactic|90|64.88|**49.98**|**54.75**|89.71|44.25|**60.71**
> PyramidKV|90|64.08|42.14|51.25|**92.98**|43.96|58.88
> AdaKV|90|**64.98**|44.92|52.38|**92.98**|**45.18**|60.09
> Quest|90|54.60|42.23|46.84|78.53|39.49|52.34
>
> We additionally evaluate Tactic on coding benchmarks, where it continues to achieve near- or even better accuracy than full attention, further demonstrating its broad applicability across domains.
>
> Method|Config|Repobench P|Lcc|Average
> -|-|-|-|-
> Full|Full|52.82|49.28|51.05
> Tactic|70|52.73|56.46|54.59
> PyramidKV|70|51.93|49.94|50.94
> AdaKV|70|50.80|49.45|50.12
> Quest|70|44.75|37.57|41.16
> ||
> Tactic|90|52.51|50.73|51.62
> PyramidKV|90|49.90|51.03|50.47
> AdaKV|90|51.30|50.32|50.81
> Quest|90|53.71|52.80|53.25
>
> ## How Does Tactic Compare To Recent Heavy-Hitter, Speculative Decoding, And KV-Eviction Methods?
> **Speculative decoding is orthogonal to our work**. In fact, Tactic's sparse attention mechanism can serve as an efficient draft model within a speculative decoding pipeline, as demonstrated in Figure 7. **Regarding KV eviction, our baselines PyramidKV and AdaKV fall into this category**. However, their performance is constrained by a fixed token budget, which limits their accuracy. Similarly, **Quest follows a heavy hitter **strategy**, but its performance suffers due to sequential token grouping and uniform budget allocation.
>
> As shown in Table 9, Tactic selects orders of magnitude fewer tokens than these baselines while achieving the same RULER accuracy, highlighting its superior efficiency.

---

> > ### Comment · Reviewer_tuQx · 2025-11-27
> > **Reply to the Authors’ Response**
> >
> > The rebuttal satisfactorily addresses several concerns, including reproducibility, architectural generality, and some level of validation at 70B scale. The multi-seed stability analysis and added long-context results materially improve the paper’s strength. However, limitations remain in real-world wall-clock comparisons against the strongest kernel-optimized base the lack of prefill acceleration for many common workloads. Overall, the paper remains a good contribution with clear practical value, and I will increase the contribution score accordingly. My overall  score is unchanged

---

### Official Review · Reviewer_Bx4T · 2025-11-01

**Soundness:** 3
**Presentation:** 3
**Contribution:** 2
**Rating:** 4
**Confidence:** 3

**Summary:**

The paper presents a novel adaptive sparse attention mechanism designed for efficient inference in long-context language models (LLMs).

Unlike existing fixed-budget methods, Tactic dynamically selects tokens based on cumulative attention scores, allowing it to adapt to variations in attention sparsity. This flexibility enhances the model's efficiency in processing long sequences.
Tactic employs clustering-based sorting and distribution fitting techniques to accurately estimate the importance of tokens while minimizing computational overhead. This approach improves the model's ability to handle large amounts of data effectively.
The experimental results demonstrate that Tactic achieves up to 7.29 times speedup in decoding attention and 1.58 times overall inference speedup compared to traditional methods, all while maintaining high accuracy.
The framework is designed to be practical for long-context LLMs, making it a valuable contribution to the field of natural language processing, particularly for applications requiring efficient handling of extensive textual data.
In summary, Tactic represents a significant advancement in adaptive attention mechanisms, providing both efficiency and accuracy for long-context language models.

**Strengths:**

1. Tactic employs a dynamic approach to token selection based on cumulative attention scores rather than a fixed budget. This adaptability allows the model to efficiently handle variations in attention sparsity across different contexts, leading to improved performance in long-context language models
2. The experimental results demonstrate that Tactic achieves up to 7.29 times speedup in decoding attention and 1.58 times overall inference speedup compared to traditional methods. This efficiency is crucial for practical applications of long-context language models, making Tactic a valuable contribution to the field
3. Despite the focus on efficiency, Tactic maintains high accuracy levels. The method's design ensures that the attention distance error is minimized, providing a theoretical guarantee on the accuracy of the attention approximation
4. Tactic utilizes clustering-based sorting and distribution fitting techniques to estimate token importance effectively. This innovative approach reduces computational overhead while ensuring that critical tokens are selected based on their relevance to the current query, enhancing the overall performance of the model

**Weaknesses:**

1. The adaptive nature of Tactic, which involves dynamic token selection based on cumulative attention scores, may introduce additional complexity in implementation compared to simpler fixed-budget methods. This complexity could pose challenges for practical deployment in certain environments.
2. Tactic relies heavily on accurately estimating attention scores for effective token selection. Any inaccuracies in this estimation could lead to suboptimal performance, potentially affecting the overall effectiveness of the model
3. While Tactic uses clustering to improve efficiency, the initial clustering step may introduce computational overhead, especially for large datasets or models. This could negate some of the efficiency gains achieved during the decoding phase
4. The performance improvements demonstrated in the paper are based on specific models (e.g., Llama-3.1-8B-Instruct). There may be concerns regarding how well Tactic generalizes to other architectures or tasks, which could limit its applicability in diverse scenarios

**Questions:**

1. Could you provide more detailed insights into the clustering methodology used in Tactic? Specifically, how do you determine the optimal number of clusters, and what criteria do you use to evaluate the effectiveness of the clustering?

2. How well does Tactic generalize to other language models beyond Llama-3.1-8B-Instruct and MegaBeam-Mistral-7B-512k? Are there any limitations observed when applying Tactic to different architectures or tasks?

3. What is the impact of different hyperparameter settings on the performance of Tactic? For instance, how does varying the average cluster size or the number of iterations in K-means affect the results?

4. Could you elaborate on the real-time performance metrics used to evaluate Tactic? How do you measure the efficiency gains during actual inference as opposed to theoretical evaluations?

---

> ### Author Response · Authors · 2025-11-26
> **Thanks for your insightful and thoughtful review. We would like to address these concerns in the following sections.**
>
> ## Does Tactic’s Dynamic Token Selection Complicate Practical Deployment?
>
> Although Tactic uses dynamic selection, we abstract this logic via optimized custom kernels and pack Tactic as **a separate module that replaces the standard attention module entirely.** This design isolates the complexity, allowing users to integrate Tactic as easily as existing sparse attention libraries without modifying the broader model architecture or deployment environment.
>
> ## How Robust Is Tactic To Attention Score Estimation Errors?
>
> Accurate attention score estimation is a fundamental dependency for all sparse attention approaches, but Tactic improves robustness through a **self-correcting design.** Instead of static estimation methods and fixed token budgets, Tactic mitigates the risk of inaccuracy by coupling high-quality token grouping with dynamic curve fitting.
> - **Curve Fitting Compensates for Grouping Errors**: If the initial token grouping is suboptimal, the resulting attention distribution exhibits a slower decay (a flatter curve). Tactic’s curve-fitting algorithm detects this shape and automatically increases the token budget.
> - **Sorting Protects Against Fitting Errors**: Even if the curve fitting yields an imperfect stopping point, Tactic’s token grouping and sorting mechanism prioritizes the most critical tokens. Because selection proceeds in order, estimation errors only affect the specific cutoff point in the tail, leaving the high-value tokens likely unaffected.
> - **Empirical Verification**: As shown in Table 3, this dual protection allows Tactic to satisfy or exceed the target cumulative score in over 85% of cases.
>
> ## How Much Does Clustering Overhead Impact Overall Performance?
>
> **The clustering phase accounts for approximately 6% of the total prefill time**, as shown in Figure 8, and shares the same scaling behavior O(n²) as the prefill step for long contexts. As a result, the relative overhead introduced by clustering remains proportionally small even as sequence lengths increase. **Even when factoring in the clustering overhead, as demonstrated in Figure 11, Tactic achieves up to 1.58× overall speedup**.
>
> ## How Well Does Tactic Generalize Across Models And Tasks?
>
> We report LongBench results on two additional models, **Qwen2 and Llama‑3.1‑70B‑Instruct**, to assess generalization. For Qwen2, at a 70% cumulative attention score target, Tactic achieves an average score of 42.82 vs. 43.53 for full attention and 40.58 for the best baseline. For Llama‑3.1‑70B‑Instruct at the same target, Tactic reaches 61.06 vs. 60.29 for full attention and 56.99 for the best baseline. These results indicate that Tactic generalizes well across architectures.
>
> We also evaluate Tactic on **coding benchmarks**, where it continues to achieve near- or even better accuracy (54.59 vs 51.05) than full attention, surpassing other baselines and further demonstrating its broad applicability across domains.
>
> **For detailed results, please refer to the response for Reviewer tuQx due to space constraints.**
>
> ## How Does Tactic Determine Clustering Hyperparameters And How Do They Affect Effectiveness?
>
> Tactic adopts K-means clustering with a single-trial random initialization (n_init=1), a maximum of 10 iterations (max_iter=10), and 16 average tokens per cluster (cluster_size=16) as its default configuration, balancing clustering quality and computational efficiency.
> To assess clustering effectiveness, Tactic uses the **Silhouette score**, a widely used metric that captures how well each token is matched to its own cluster compared to other clusters. A higher Silhouette score indicates tighter intra-cluster cohesion and clearer inter-cluster separation—both desirable for reducing attention redundancy during decoding.
>
> We conducted a comprehensive ablation study (Table 4 in the paper, also shown below) to explore how clustering quality varies with key hyperparameters:
>
> |Hyperparameter|Value|8192|16384|32768|65536
> |-|-|-|-|-|-
> |n_init|1|0.091|0.113|0.145|0.166
> ||2|0.095|0.117|0.150|0.170
> ||4|0.095|0.117|0.149|0.170
> ||8|0.095|0.118|0.149|0.170
> |max_iter|5|0.091|0.111|0.145|0.166
> ||10|0.091|0.113|0.144|0.165
> ||20|0.095|0.117|0.150|0.171
> ||30|0.095|0.117|0.150|0.171
> ||40|0.095|0.117|0.150|0.171
> ||50|0.095|0.117|0.150|0.171
> |cluster_size|8|0.096|0.118|0.155|0.173
> ||16|0.095|0.116|0.149|0.170
> ||32|0.088|0.110|0.143|0.164
> ||64|0.082|0.104|0.131|0.155
> ||128|0.070|0.096|0.119|0.143
>
> ## How Are Tactic’s Real-Time Inference Efficiency Gains Measured?
>
> In our efficiency evaluation, we measure both attention module latency and end-to-end runtime as key performance metrics. To capture actual inference behavior, we run the inference framework including our custom kernels, and **use NVTX markers and CUDA events for fine-grained timing breakdowns**. These measurements directly reflect the **real-time inference performance** rather than relying on theoretical estimates.

---

### Official Review · Reviewer_zjbE · 2025-11-11

**Soundness:** 2
**Presentation:** 2
**Contribution:** 2
**Rating:** 4
**Confidence:** 4

**Summary:**

This paper aims to address critical inefficiencies in long-context LLM inference by proposing a calibration-free sparse attention mechanism that dynamically selects tokens based on cumulative attention scores to meet a target fraction, rather than relying on a fixed token count.

**Strengths:**

1. The paper is well structured and easy to follow.
2. This paper addresses a real, practical problem, and this area is promising.
3. The evaluation is comprehensive.

**Weaknesses:**

1. Vague practical cost tradeoffs: Clustering overhead is <6% of prefill time, but no breakdown of how this scales with ultra-long sequences (>128K) or batch inference—critical for real-world high-throughput serving.
2. The idea is similar to several other papers, such as Twilight: Adaptive Attention Sparsity with Hierarchical Top-p Pruning.
3. Typo: line 43, reference

**Questions:**

1. Any ablation study on distribution fitting? Compare \(y=\frac{a}{x}+b\) with power-law and analyze the fitting error impact on results.
2. The selector will cause any latency overhead; some papers have mentioned this [1]. Can you add some analysis?

[1] HShare: Fast LLM Decoding by Hierarchical Key-Value Sharing.

---

> ### Author Response · Authors · 2025-11-26
> **Thanks for your insightful and thoughtful review. We would like to address these concerns in the following sections.**
>
> ## What is the cost of Prefill for longer sequences and batch inference?
> **Clustering overhead remains consistent relative to prefill time for ultra-long sequences and batched inference.** Apart from the 32-128K results, we profiled a 256K-token sequence where clustering required 11.37 s, which is 6.6% of the 170.75 s prefill time. Because both prefill (self-attention) and clustering (distance computation) scale quadratically with sequence length, their asymptotic ratio remains constant beyond 128K tokens. Similarly, for batched inference, applying clustering independently to each sequence ensures both costs scale linearly with batch size, preserving the <7% overhead ratio in high-throughput settings.
>
> ## How does Tactic compare to Twilight?
>
> **Twilight is concurrent work that shares our accuracy goals but suffers from higher costs due to its hierarchical design.** Twilight employs a complex, two-stage pipeline: it first uses an off-the-shelf top-K algorithm (e.g., Quest) to filter candidates, then re-estimates per-token scores for the survivors using the dot product between the query and a 4-bit quantized key cache. Based on the score, Twilight applies a final top-p selection and performs sparse attention on those tokens. Compared to Tactic, Twilight falls short on the following aspects:
>
> - **Latency**: Twilight consists of two steps that first perform top K, then perform top P, thus having higher latency. The second step of Twilight (score estimation and top-p) takes $2\times$ the time of the initial filtering and is an order of magnitude slower than the sparse attention operation itself, according to their evaluation. Tactic features lower overhead (around 10% of the full attention) via a unified, single-stage selection.
> - **Memory**: Twilight’s storage overhead is additive. It retains the metadata required by the first-stage filtering algorithm (e.g., Min/Max vectors for Quest, which are a similar size to Tactic’s cluster centroids) and requires storing the additional 4-bit quantized key cache for the second stage, which further increases the total KV cache footprint by 12.5%, limiting the max sequence length and batch size in throughput-oriented serving.
> - **Compatibility**: Twilight’s reliance on a specific auxiliary 4-bit cache limits compatibility. If the base model is already running quantized inference (e.g., 4-bit or 8-bit), it might be infeasible to further compress their 4-bit cache to 2 or 1 bit. Thus, maintaining Twilight's separate structures for quantized inference would incur even larger overhead. However, Tactic can naturally use the same data format as the main model for cluster centroids, proportionally reducing the overhead.
> - **Dependency**: Twilight requires existing top-K methods as a pre-processing step, which sets an accuracy upper bound. Additionally, it is unclear how to tune the threshold for the first stage properly for various models and tasks.
>
> ## How Does The Choice Of Distribution Fit (a/x+b vs Power-Law) Affect Results?
>
> We report the accuracy on LongBench alongside the number of tokens selected by each curve fitting algorithm. **Besides const, all other curves model the monotonically decreasing trend of attention scores and achieve comparable accuracy.** Among them, the inverse function selects the fewest tokens while maintaining high accuracy, making it the most efficient choice.
>
> Fit Algo|Threshold|Hotpotqa Acc|Hotpotqa Tok|Multifieldqa_en Acc|Multifieldqa_en Tok|Musique Acc|Musique Tok
> -|-|-|-|-|-|-|-
> Inverse|70|**56.89**|**719**|52.81|**459**|29.87|**878**
> Linear|70|55.90|764|52.96|499|29.67|944
> Exp|70|**56.89**|761|**53.49**|496|**30.09**|928
> Quadratic|70|56.17|776|52.98|508|29.88|953
> Const|70|55.84|11072|52.46|6561|27.81|13236
> ||
> Inverse|90|**57.17**|**1707**|54.19|**1251**|**29.57**|**2056**
> Linear|90|56.67|1812|53.52|1335|29.54|2203
> Exp|90|56.67|1769|**54.26**|1293|29.46|2142
> Quadratic|90|56.67|1884|53.74|1387|29.54|2287
> Const|90|56.67|12661|52.46|7478|28.21|15148
>
> Fit Algo|Threshold|Narrativeqa Acc|Narrativeqa Tok|Qasper Acc|Qasper Tok|Triviaqa Acc|Triviaqa Tok
> -|-|-|-|-|-|-|-
> Inverse|70|29.34|**1417**|44.74|**336**|89.46|**599**
> Linear|70|29.28|1465|45.28|377|88.71|620
> Exp|70|29.35|1501|45.15|369|89.16|621
> Quadratic|70|**29.85**|1518|44.66|379|89.76|636
> Const|70|29.05|20010|**45.49**|4393|**90.26**|9852
> ||
> Inverse|90|29.41|**3460**|44.69|**990**|90.30|**1356**
> Linear|90|**30.38**|3740|44.80|1051|**90.80**|1397
> Exp|90|29.43|3603|45.28|1003|**90.80**|1381
> Quadratic|90|30.18|3863|44.69|1099|90.40|1463
> Const|90|29.25|23011|**45.32**|4987|90.26|11319
>
>
> ## Does The Selector Introduce Significant Latency Overhead?
>
> Through kernel optimizations, we limit the selector's latency overhead to ensure net performance gains. **As shown in Figure 9, the combined sorting and curve fitting requires approximately 10% of the attention time for long sequences.** Despite this cost, Tactic achieves a net speedup of up to $7.29\times$ compared to full attention.

---

### Comment · Area_Chair_3Jua · 2025-11-23
**Reviewer & Author Discussion**

Hi Reviewers,

Please kinly and actively participate in the review-author dicussion, raise your further concerns so that the authors can explain more, and make your final decisions.

---

### Meta-Review · Area_Chair_9Qxe · 2026-01-12

**Summary:**

Reviewers agree the paper targets a real serving bottleneck for long-context decoding and that the central idea—selecting tokens to hit a target fraction of cumulative attention mass rather than a fixed budget—is sensible and practically motivated. The main reasons the paper did not receive stronger enthusiasm are about (i) whether the practical tradeoffs are fully nailed down for real deployments, especially clustering/selector overhead under very long contexts and throughput settings, and (ii) positioning versus concurrent/related adaptive sparsity methods (e.g., Twilight) and strong production baselines, including kernel-optimized attention and other decode-time accelerations. Two reviewers below-threshold also raised concerns that the method may introduce implementation complexity and that the evaluation might not convincingly establish generality beyond the few model families initially emphasized, as well as asking for clearer ablations to disentangle the roles of clustering versus score-based selection.

**Reviewer Concerns:**

The rebuttal is generally responsive on the concrete technical questions. It provides additional profiling showing clustering remains a small fraction of prefill time even at 256K context and argues the overhead ratio is stable because both prefill and clustering scale similarly; it also adds analysis of selector latency (sorting + curve fitting) as a small fraction of attention time and includes a targeted ablation separating clustering quality from adaptive allocation (with “Tactic-topK” substantially degrading accuracy at the same token counts, and “position-cluster” requiring more tokens for similar accuracy). It also expands generalization evidence to additional models including a 70B model and includes multi-seed stability analysis; taken together, these directly address several “missing evidence” style concerns. What remains less fully settled is the strongest version of the “real-world” comparison: the discussion acknowledges that wall-clock comparisons against the best kernel-optimized full-attention implementations and end-to-end production stacks remain limited, and the method still does not accelerate prefill, which can matter for many common workloads. The novelty/positioning concern relative to Twilight is addressed at a high level (latency/memory/compatibility arguments), but since Twilight is concurrent work, the paper’s distinctiveness will still read as incremental to some readers unless the final version is especially careful and precise in framing.

**Reviewer Scores:**

Reviewer tuQx (score 6) participated in the discussion and stated that the rebuttal satisfactorily addressed reproducibility, architectural generality, and some validation at 70B scale, and that they would increase the contribution subscore while keeping the overall score unchanged; given their starting point and explicit comment, the AC expects no overall score change.

Reviewer gwJo (score 6) asked for an ablation isolating clustering and the adaptive selection mechanism; the rebuttal provides exactly this kind of decomposition with clear evidence that naive allocation across heads/layers substantially harms accuracy and that weaker clustering increases token needs. gwJo did not participate further in the discussion, so the AC will not attribute intent; under a conservative read, this likely resolves the key open question and would make the reviewer more comfortable at the same 6, rather than increasing the score.

Reviewer zjbE (score 4) raised practical cost tradeoffs (ultra-long context and batch), similarity to Twilight, and requested distribution-fitting ablations and selector overhead analysis; the rebuttal addresses each with additional profiling at 256K, a concrete comparison narrative to Twilight, a table comparing fitting choices (showing “inverse” achieves similar accuracy with fewer tokens than other fits, and “const” essentially degenerates to near-full selection), and an explicit selector overhead statement. zjbE did not participate further; given their “would not mind if accepted” stance, the AC thinks the rebuttal would plausibly move them to a borderline accept, but because their initial score is below threshold and we lack a confirmation, the AC will treat this as a plausible but not guaranteed shift.

Reviewer Bx4T (score 4) raised more generic concerns about implementation complexity, robustness to score estimation, clustering overhead, and generalization; the rebuttal responds with integration details (module replacement), a robustness argument (curve fitting reacts to flatter decay by selecting more tokens) plus an empirical claim that the target cumulative score is met in most cases, and added cross-model results including 70B. Bx4T also did not participate further; here as well, the rebuttal likely alleviates many concerns, but the review itself reads somewhat template-like and not tightly tied to specific failure modes, so the AC would expect at most a modest upward correction if the reviewer were to re-evaluate carefully.

---

### Decision · Program_Chairs · 2026-01-26

Accept (Poster)